# Imaging Based Techniques Combined with Color Measurements for the Enhancement of Medieval Wall Paintings in the Framework of EHEM Project

**DOI:** 10.3390/jimaging10070159

**Published:** 2024-06-29

**Authors:** Paola Pogliani, Claudia Pelosi, Luca Lanteri, Giulia Bordi

**Affiliations:** 1DIBAF Department, University of Tuscia, Largo dell’Università, 01100 Viterbo, Italy; pogliani@unitus.it; 2DEIM Department, University of Tuscia, Largo dell’Università, 01100 Viterbo, Italy; llanteri@unitus.it; 3Department of Humanities, University of Rome 3, Via Ostiense 234, 00146 Roma, Italy; giulia.bordi@uniroma3.it

**Keywords:** medieval wall paintings, colorimetry, hypercolorimetric multispectral imaging, color reconstruction

## Abstract

(1) Background: This paper illustrates an innovative methodological approach chosen to study and map the colors of the medieval wall painting of Santa Maria Antiqua in the Roman Forum, one of the pilot sites of the EHEM project (Enhancement of Heritage Experiences: The Middle Ages). Digital Layered Models of Architecture and Mural Paintings over Time). (2) Methods: Two methods were employed to gather information about colors and mapping. Specifically, colorimetry was utilized for spot measurements, and hypercolorimetric multispectral imaging (HMI) was employed to map the same colors sampled through colorimetry. (3) Results: Chromatic data for all colors in the wall paintings were obtained in the CIELAB color space. Additionally, chromatic similarity maps were generated using the innovative HMI system, a multispectral imaging technique capable of obtaining color data information through advanced calibration software named SpectraPick^®^ (Version 1.1). This comprehensive approach facilitates a thorough understanding of color characteristics and distribution. (4) Conclusions: The color measurements and mapping represent significant advancements in the interpretation of medieval wall paintings, which are often fragmentary and stratigraphically complex. This research sheds new light on the colors used and enhances our understanding of the original appearance of the iconographic patterns. Furthermore, it enables the reconstruction of colors that closely resemble the originals.

## 1. Introduction

The research presented in this paper has been developed within the project EHEM, Enhancement of Heritage Experiences: The Middle Ages. Digital Layered Models of Architecture and Mural Paintings over Time, JPI Cultural Heritage—Horizon 2020 Conservation, Protection and Use. To develop the project, a consortium between the Universitat de Barcelona, the University of Rome 3, the University of Tuscia and the Technological Center CYENS in Nicosia (Cyprus) was created.

The main aim of the project was to face the complexity of the medieval monuments by investigating three pilot sites: Sant Quirze de Pedret (Spain); the Enkleistra of Agios Neophytos (Cyprus); and Santa Maria Antiqua (Rome), with a specific focus on combining technical instruments and methodological approaches able to improve the knowledge of the monuments and, above all, the fruition quality for various audiences. The EHEM project aims to create a “Digital Twin” of each selected pilot site. This digital representation of the real monument integrates information from different sources, visualizes the monument through the various transformations it has undergone over the centuries, and allows for simulations and analyses [1].

This process is particularly relevant in medieval monuments because they often contain very complex, articulated and fragmentary wall paintings, such as in the case of the church of Santa Maria Antiqua in the Roman Forum. That is the specific object on which the present paper will focus attention.

The church of Santa Maria Antiqua was discovered in 1900 on the slopes of Palatine. The church was abandoned in the ninth century, probably after an important earthquake that affected the area of the Roman Forum. The Santa Maria Antiqua church, which was established in the 6th century within the structures of the Domitian age, preserves mural paintings on the walls dating back to between the 6th and 11th centuries, mostly superimposed on each other in a palimpsest. Overall, there are around twenty decorative phases, with the fulcrum of early medieval pictorial evidence found on the wall to the right of the apse (the so-called “palimpsest wall”), which features eight layers of wall paintings executed between the 4th and 8th centuries [2].

The indisputable quality of the paintings of Santa Maria Antiqua is today reflected in the state of conservation in which they came to us: fragmentary, incomplete and with a weakened pictorial film, as well as a lack of surface finishes. The state of conservation is the result of the paintings being buried until their discovery and the subsequent changes they underwent, particularly the whitening of the surface when exposed to air and high relative humidity. Furthermore, during the restorations carried out between 1900 and 1910, substances such as wax and glue were applied to the surface, which left traces on the pictorial surface, altering the perception of colors [3]. This is a substantial aspect when, as in our case, the painted surfaces must be digitally replicated to create their “digital twin” to be used as a dissemination tool.

In the EHEM project, the role of color in wall paintings was decisive. This is why we set out to acquire the specific physical and chemical characteristics of the colors by combining them with the characteristics of the constituent materials and the manner of execution. From a methodological point of view, together with the autoptic investigation and the analytical–scientific characterization of the pigments performed in previous work [4] aimed at recomposing the palette used by the painters who worked in Santa Maria Antiqua, multispectral and colorimetric investigations were prepared to implement knowledge of the color characteristics of the paintings as they appear today.

The multispectral reading of medieval wall paintings in a compromised state of conservation, such as the one present in Santa Maria Antiqua, allows the image to be “re-composed” and enhanced where finishes, highlights or brushstrokes are no longer perceivable in visible light or have weakened. A similar investigation was conducted on Giotto’s paintings, although with a different approach [5].

Here, we propose, for the first time, the combined use of an innovative technique named hypercolorimetric multispectral imaging (HMI), coupled with colorimetry, for mapping the colors, thus allowing the images to be “returned” for their digital reproduction. Multispectral imaging investigations enable us to deepen our knowledge of the execution technique and the state of conservation of the paintings while also adding valuable information for defining color, especially where it is present only in traces. As demonstrated in the literature, imaging techniques have become widely applied in cultural heritage because they can provide a complete knowledge of the surfaces in a fast and reliable way without the need for sampling micro-chips of materials for laboratory analysis, a fact that is in general highly desirable, or even mandatory in some cases [6,7,8,9,10,11,12,13]. In our approach, the potentiality of multispectral imaging was exploited to obtain the chromatic distribution of the colors accurately measured by point colorimetry applied to some selected areas of the wall paintings covering the most relevant historical phases of the church decoration. HMI is a multispectral imaging technique recently developed by the society Profilocolore (Rome, Italy), whose principle and potential application have been reported in some published papers [14,15,16,17]. This technique demonstrated its great utility in the investigation of paintings and other kinds of artwork [18,19,20,21].

Color measurements are widely used in cultural heritage to perform a quantitative evaluation of changes due to various kinds of treatments, such as cleaning, the application of protective substances and consolidants, the use of inpainting, etc. [22,23,24,25] and for documentation purposes [26,27,28,29,30]. Color can be measured by means of different kinds of colorimeters and spectrocolorimeters, but also through software specifically developed to derive color information from paintings or, in some cases, from photographs of an artwork if, for example, the original is no longer available [31,32,33]. An interesting approach to the investigation and mapping of color was proposed by some authors in order to obtain a diagnosis of the conservation status of a painting [34]. In this case, color measurements, by means of a colorimeter and a digital image analysis program, were used to evaluate the damage area on the investigated painting [34].

Another application of color measurements has been proposed by Liang and Wan for the purpose of digital conservation of ancient wall paintings using a prototype pigment color chart [35].

Regardless of the systems and methodologies adopted, color measurement in cultural heritage is of fundamental importance for numerous applications, and some papers highlight its limitations and advantages [36,37].

In the present paper, we report the results, in terms of color similarity maps and chromatic coordinates, of the Santa Maria Antiqua wall paintings. The decorative arrangement and the methodologies used to gather color information are described in Section 2; the discussion aims to comment on the results and compare them to previous data on pigment analysis in order to summarize the main findings of the study and propose possible future developments of the research.

## 2. Materials and Methods

The methodology proposed in this paper and used to support the EHEM project, was based on the combination of an imaging technique and a spot measurement. The first allows for the mapping of the different colors visible in the different areas chosen for the application of the method; the second (i.e., spot color measurement) gives precise data for each color.

So, by combining the maps and the chromatic coordinates, we can obtain a complete palette of the paintings.

The used techniques are hypercolorimetric multispectral imaging (HMI) and colorimetry by a portable digital instrument, as detailed in the following sub-paragraphs, after having briefly described the painted areas chosen for the measurements.

### 2.1. Selected Paintings and Mapping Methodology

One of the crucial issues one faces when studying the wall paintings of Santa Maria Antiqua is the complex stratigraphy of the ten pictorial phases in the Presbyterian area. In the EHEM project, in order to understand the use of colors in the different phases, eight representative sample areas were selected (Appendix A).

The numbering of the areas follows the order of acquisition of the images, which were selected by favoring the painted portions accessible without scaffolding but representative of the diachrony of the paintings and their stylistic characteristics. The images provide information on the use of colors between the 6th and 9th centuries by local workshops of painters and others influenced by Hellenistic modes or Byzantium.

Following the chronological order, the first area is the image of St. Anne holding the Child Mary in her arms, painted on the right wall of the presbytery and dated to the last quarter of the 6th century (area 2, Appendix A). This is followed by a nucleus of paintings datable to the 7th century that belong to different decorative phases and, thus, to different workshops. These are the paintings on the small wall to the left of the apse where a false marble area was painted in the first decade of the 7th century (area 4A, lower part, Appendix A) and later, in the upper part, the figures of the Church Fathers pertaining to the year 663 AD (area 4B, upper part, Appendix A). On the pillars of the nave, in the first half of the 7th century, various scenes were painted. For the color measurements, the Solomon and the Maccabees scene was chosen. This scene was painted on the lower part of the northern side of the pillar positioned in the south-west area of the nave (area 6, Appendix A).

At the time of Pope John VII (705–707), the decoration of the church, specifically in the area of the chancel, was completely renewed, and various scenes were added to the pillars of the nave. To compose the palette used by the painters, our attention went to three areas: the high choir (area 1, Appendix A), the veil painted on the left wall of the Presbytery (area 3, Appendix A) and the scene with the Annunciation (area 7, Appendix A).

A homogenous core is represented by the paintings in the Theodotus Chapel, created at the time of Pope Zechariah (741–752), which opens to the left of the chancel. Here, as a reference area for color investigation, the area depicting the so-called ‘family portrait’ with the donors next to the figure of the Virgin was investigated (area 5, Appendix A).

The decorative phases of the church are completed by the pictorial arrangement created at the time of Pope Stephen II (752–757). From this phase, we have taken into consideration a portion of the left wall of the eastern side with the theory that contains depictions of figures of saints and the fathers of the Western and Eastern Churches (area 8, Appendix A).

### 2.2. Colorimetry

Color measurements were performed on selected points in order to cover the entire palette of the paintings. The color measurement points were carefully documented in detail and in the general photograph of each selected area (see Appendix A). An EOPTIS (Trento, Italy) CLM-194 colorimeter was used for the color measurements. It is a handheld compact device that can be used for measurements on the surface of a wide range of matte and glossy materials, thanks to a 45°/0° fixed geometry and the very uniform illumination provided by integrated LEDs. The device is powered and interfaced through a USB port using a laptop with the Microsoft Windows operating system. The used instrument was specifically equipped with a conical head, allowing for the measurement of a spot 6 mm in diameter. The standard observer 10° and the illuminant D65 were selected for the measurements.

Before each set of measurements, the instrument was calibrated with a white reference standard supplied by EOPTIS. This white was also used as a reference for the color measurements.

The CIEL*a*b* color space was used, where L* represents lightness and ranges from 0 (black) to 100 (white), and a* and b* are the chromatic coordinates that can assume positive and negative values. The coordinate a* indicates the red (positive–green (negative) axis, whereas the b* coordinate indicates the yellow (positive)–blue (negative) axis.

Three measurements were performed at each selected point. The average values were then calculated and reported in the figures shown in the result section.

### 2.3. HMI Technique

The HMI system and procedure have been explained in previous papers, so here a synthesis of the main characteristics and working methodology is summarized [14,19].

First of all, to obtain the calibrated multispectral images, the acquisition is performed through a Nikon (Nital SpA Moncalieri Torino, Moncalieri, Italy) D800FR (Full Range) camera, a 36=megapixel reflex camera modified under a Nikon (Nital SpA Moncalieri Torino, Italy)/Profilocolore^®^ (Rome, Italy) common project to achieve an extended 300–1000 nm range sensitivity, with the use of portable camera flashes also modified in full range. The accurate calibration of the system is carried out through a color checker made of 36 color patches from the NCS (Natural Colour System^®^, Milan, Italy) standard catalog and the use of white reference patches with 98% spectral reflectance. The spectral reflectance of the color checker and white references was measured through a laboratory spectroradiometer (Instrument System Spectroradiometer CAS 140 CT, Instrument Systems, Munich, Germany) in the range 220–1050 nm with 0.7 nm accuracy in a dark room in the Profilocolore^®^ (Rome, Italy) laboratory.

Image acquisition in the UV-NIR range is performed using two bandpass filters called A (whose spectral transmittance includes the UV and VIS range) and B (including the VIS and NIR bands until 1000 nm) designed by Profilocolore^®^ (Rome, Italy). The transmission spectra of the two filters are published in [20].

A great advantage of HMI is the short time needed for the acquisition—only two shots are needed—without the necessity of any power supply, thus providing a powerful instrument for in situ investigation, such as in the case of Santa Maria Antiqua.

The acquired raw images are then calibrated through the HMI software SpectraPick^®^ (Version 1.1), which produces seven monochromatic images in tiff format centered at 350, 450, 550, 650, 750, 850 and 950 nm, and the RGB output. The folder with the images produced after calibration can be processed through the software PickViewer^®^ (Version 1.0), included in the HMI system, that is used to apply different kinds of tools to the calibrated images in order to extract information useful for the analysis and knowledge of the artwork.

Specifically, PickViewer^®^ (Version 1.0) was used to calculate the chromatic similarity maps of each selected color visible on the wall paintings. To do this, after having selected a color, which is the same as measured by the colorimeter, the software was asked to find all pixels in the image having similar chromatic values.

An example of the procedure followed for obtaining the chromatic similarity maps is displayed in Figure 1.

## 3. Results

The results of the HMI mapping and colorimetry were synthesized for each acquired area of the Santa Maria Antiqua wall paintings. In Figure 2, the results obtained for area 1, with the scene of David and Goliath (high choir), are shown.

In the first examined area, seven main colors have been identified. Point 1 in Figure 2 refers to the light green background whose average chromatic coordinates are: L* = 59.6, a* = −7.40 and b* = 12.2. From the map of the chromatic similarity, the green color is distributed in the background of the area (white pixels in the map) and not in other parts of the painting.

Point 2, which is a red color, has been used in a limited portion of the painting and is well defined in the map of chromatic similarity. Point 3 is a dark green used in the background, superimposed on the light green selected in point 1; for this reason, the distribution of this color is also in the area where the light green has been mapped. Point 4 is a yellow color that shows a well-defined distribution in the painting elements, being mapped in the decorative elements on the left, in the legs of the figure in the center, in the ground, in the frame (lower part), and in the flourish of the scarf.

Point 6 is taken from a white area of the dress of the figure in the center of the scene. The white color is distributed on almost all the surfaces, apart from the yellow zones.

Finally, the black color was chosen in correspondence with points 5 and 7. From the maps, it can be seen that this color is distributed throughout the surface, apart from the light yellow areas that characterize the two hills in the lower part of the background and the flourish.

The second examined wall painting was that representing St. Anne with the Child Mary, located on the right wall with respect to the central apse of the church (Figure 3).

In this case, we also obtained the color mapping associated with the values of the chromatic coordinates in the selected areas. It is interesting to note that some colors are limited to a few areas of the painting, such as the yellow color (point 4) that has been mapped only in the halo of the saint, or the light blue color that is present in the frame with the inscription, in the veil of St. Anne and in the garment of the Child Mary (point 6 and point 1).

The green color visible in the area of point 2 does not exhibit the coordinates of this color (a* should have a negative value). Instead, from the value of b*, it can be deduced that it has a yellow hue. This leads us to suppose that the green color was obtained by mixing pigments able to produce such a hue.

Point 7, corresponding to a white area, has been mapped in the entire area, apart from the yellow, red and blue zones; even that referred to point 1, which is a slightly bluish white. This led to assuming a different mixture for obtaining the white in the garment and that in the frame with the inscription. Considering the similarity between point 1 and point 6 colors, it may be derived that the difference with the white in point 7 is due to the presence (in points 1 and 6) of a blue pigment or a pigment having a bluish hue, such as, for example, vine black. This last black pigment, in fact, gives a clear bluish hue to the wall painting if mixed with lime white, and is frequently used due to this characteristic [38].

Point 5 corresponds to a dark red color used for the figures’ contours and, probably mixed with the white, to obtain the pink color of the Virgin mantle.

Area 3, representing aniconic decoration (Figure 4), has a few basic colors: yellow (point 1), red (point 2), bluish gray (point 3), black (point 4) and white (point 5).

The chromatic similarity maps show a net distribution of yellow and red colors, indicating the probable presence of pure pigments; on the other hand, the black and white colors (points 4 and 5) exhibit chromatic characteristics similar on the entire surface, apart from the yellow and red areas where the colors are recognized as completely different. This is probably due to the presence of a white pigment mixed with the black or the whitening of the surface in correspondence with point 4. In fact, the wall surfaces of Santa Maria Antiqua are whitening due to the high values of relative humidity in the church that caused recrystallisation of calcium carbonate [4]. This problem was partially solved with past and recent structural interventions, but the environmental conditions are not completely stabilized yet [39].

Area 4 is characterized by two different pictorial phases superimposed on each other: the false marble decoration, visible in the lower part of the area, and the 663 AD phase, visible in the upper part of the area (Figure 5 and Figure 6 and Appendix A).

In this area, color measurement points from 1 to 8 refer to the false marble decoration; points 9 to 13 are taken from the layer dating back to 663 AD (the pictorial phase realized during the Pontificate of Vitalian). This layer was very fragmentary, and it was not possible to reach the upper parts, so only five points were measured and mapped, but they were representative of the palette of the 663 AD phase.

In the false marble decoration, the various colors are generally well-defined in the map, such as brown (point 1), yellow (point 2), red (point 4), black and different grays (points 3, 5, 7 and 8).

Point 6 is a thin white line (the sole white zone where it was possible to take a color measurement). Apart from the yellow zones, white is present in all the other colors and in the grouting, according to the obtained mapping. In medieval wall paintings, white is usually obtained by using calcium carbonate from various sources, which is also the binder of the pigments in the fresco technique, so it is not strange to find this white combined with the other colors [40].

The painting referring to the 663 AD phase was sampled at five points: yellow-orange (point 9), black (point 10), red (point 11), yellow (point 12) and white (point 13). Red and yellow colors are well-defined, whereas white is distributed on the entire surface, apart from the red and yellow areas, being probably mixed with the other pigments to obtain the light hues visible on the painting of the 663 AD phase.

Area 5 has been chosen in the so-called Theodotus Chapel, which is located on the left side of the church with respect to the central apse. The results of the color measurements and mapping are shown in Figure 7.

The main visible colors are blue (point 1), yellow (point 2), yellow-orange (point 3), white (point 4), brown (point 5), black (point 6), red (point 7) and green (point 8). Blue is concentrated in the garment of the central figure (point 1).

Two kinds of yellow have been measured: a lighter hue in point 2 distributed in the pavement and in the garments of the first and third figures; and a darker yellow mapped in the footstool and in the upper part of the first figure’s garment (point 3).

The white color has been sampled in one of the pearls that decorates the footstool, where this color appears more intense (point 4), and in the mapping, it is concentrated in the garment of the first figure and in the mantle of the fourth one. White is also mapped in the lines of the frame and in the garment of the central figure, probably added to the blue to create highlights in the folds. Clearly, the white color has been mapped further in the lacunae, and grouting is visible in various zones of the panting.

Points 5 and 7 are associated with brown and red colors used for the garments, for the hair of the second figure (from the left), for the contours of the garment, for the frame dark lines, and for the footstool base. It must be noted that the value of the a* coordinate in point 5 is negative, suggesting the possibility that the colorimeter spot collected not only the brown of the garment contour but also part of the green background due to the thin contour thickness. So, this measure cannot be considered correct. On the other hand, the map that is obtained from a much smaller area of the image refers exclusively to the brown contour. This result further demonstrated the relevance of hypercolorimetric multispectral imaging in the in-depth investigation of colors’ distribution.

Point 6 refers to a black line defining the frame of the painting. The black is also mapped in the background and in the garment of the central figure, probably because a black pigment has been added to the painting materials to obtain a more saturated and dark color. This is the usual approach in Byzantine wall paintings [41].

Lastly, the green color characterizes the background of the area, as visible in Figure 7 (mapping point 8). The chromatic coordinates identify a real green color (a* has a negative value), and the mapping clearly defines the distribution in the background of the scene.

The next examined area is number 6, located in one of the pillars on the right side of the church, representing Solomon and the Maccabees (Figure 8).

In this area, we sampled eight points corresponding to the main colors visible on the paintings: dark and light gray bluish (1 and 2, respectively), yellow (point 3), green (point 4), dark red (point 5), black (point 6), brown (point 7) and white (point 8).

The white and gray areas have color similarities, as found in the mapping of points 1, 2 and 8. Yellow, red and brown colors have defined areas and appear to have been used pure, without mixing white or black. Moreover, points 5 and 7 have very similar maps, suggesting the use of the same pigments. The black color, sampled in point 6, was also recognized in the gray and light gray areas, including lacunae and grouting, probably due to the addition of a black pigment.

Another painting chosen to apply our methodology for sampling colors is area 7, attributed to the phase of Pope John VII (Figure 9).

This area represents the Annunciation of the Archangel Gabriel to the Virgin Mary. It is a detached area positioned on a mobile support, close to the entrance to the church.

Points 1 and 6 refer to yellow colors that, from the mapping and the chromatic coordinates, are found to be similar. Point 5 is another yellow that shows different coordinates, especially b*, which has a higher value, indicating a more saturated color. Point 2 is a dark pink color applied over the yellow, and in fact, it is also distributed in the background on the right of the Archangel. Point 3 is a greenish color whose mapping is limited to the line in correspondence with the Archangel’s garment and the frame on the left. Some similarities have also been found in the background, probably due to the presence of common pigments or other materials (surface whitening due to calcium carbonate, as previously written). Point 4 is the dark pink color of the background that is well-highlighted on the right side of the area. Point 7 is on the white color of the Archangel’s dress. It is present in the white parts of the painting but further in the grouting and lacunae, where calcium carbonate may be hypothesized as the main constituent material.

Points 8, 9, 10 and 11 refer to dark colors: brown, dark red and black.

The maps show similarities between points 8, 9 and 4, probably because the pink color was obtained with the same materials but with a different combination of white and black added to darken or lighten the hue of the red pigment.

Point 10 is a completely different color, distributed only in the brown areas of the painting, probably associated with a brown earth.

The last examined area is that of Pope Stephen II (Figure 10). The seven selected points represent the man’s color observed on the painting: white (point 1), yellow (point 2), black-gray of the dress (point 3), red (point 4), black of the frame (point 5), light yellow (point 6) and another red (point 7).

The white color (point 1) is clearly well-distributed in the white parts of the panting, but also in the lacunae and in the black-gray zones, which are probably obtained by mixing black and white pigments. Point 2 is a yellow color distributed not only in the garments but also in the halo of the first saint from the right. The other yellow (point 6) is found in the same areas as point 2 mapping, but further in other parts of the dresses and in the pavement, where it was sampled for the analysis. From the values of the chromatic coordinates, the yellow referred to in point 2 is more saturated with respect to that measured in point 6, where a mixture of yellow and red pigments seems probable. Points 4 and 7 are associated with red colors. The one sampled in point 4 is well-distributed in the red areas of the garments, in the background and in the upper frame. The red corresponding to point 7 has similar mapping to point 4, but it appears concentrated in some areas: the lines used to define the wrinkles on the lapel of the garment, the upper frame and the lower part of the background right side.

## 4. Discussion

In the church of Santa Maria Antiqua, a complex wall painting arrangement is visible today, after several restoration campaigns subsequent to the discovery and excavations that brought this incredible pictorial palimpsest to light.

The wall paintings were made in different historical periods and by various painters, sometimes overlapping several layers, such as on the right wall of the central apse, where the so-called *Angelo Bello* (The Beautiful Angel) can be admired; for a detailed description of the wall palimpsests, see [42,43,44,45].

During the long restoration that affected the wall paintings of Santa Maria Antiqua, various diagnostic campaigns were always performed with spot analysis of pigments and binders that allowed for the characterization of most of the constituent materials [4,46,47,48].

However, until now, a comprehensive color measurement campaign had never been performed. For this reason, the EHEM project focused on the evaluation and mapping of colors.

The main colors observed on the wall paintings, in the different areas and historical phases, can be grouped into eight main hues: blue, gray, green, yellow, red, brown, black and white.

### 4.1. Blue and Grey Hues

Rarely a true-blue color has been detected, but rather a gray-bluish appearance was found, such as in St. Anne point 6 (Figure 3), area 3 point 3 (Figure 4) and the Maccabees points 1 and 2 (Figure 8). The only true-blue color is in point 1 in the Theodotus chapel painting (Figure 7). Analyses of various samples of blue colors or colors bluish in appearance have been performed in the past and revealed two kinds of materials used for obtaining the blue appearance: Egyptian blue and wine black [4,46,47,48]. This last black pigment is characterized by a bluish hue, but the chromatic coordinates indicate that it is a carbon-based pigment, not a real blue. Otherwise, the Egyptian blue is characterized by a* and b* coordinates typical of this pigment, in particular a pale grayish blue, as reported by Bianchetti et al. [49].

### 4.2. Green Hue

Green colors have generally been obtained by using green earth, specifically celadonite [4,47,50]. However, these are not always pure colors but are sometimes made by mixing different pigments to obtain variations in hue.

The study of the constituent materials has revealed the widespread use of earth green pigment, which is often darkened with carbon black or lightened with lime white. Additionally, there is evidence of a less common green hue obtained by mixing yellow ochre with carbon black (Figure 11). It is a specific color created by the mixture of two non-green pigments to obtain a pastel hue, which marks the work of a specific workshop responsible for the paintings of the so-called phase of *Angelo Bello* on the palimpsest wall and on the area with Saint Anne and Child Mary painted on the west wall of the presbytery [48]. In the post-production of the photographic images of these paintings for the digital model, we could also attribute the term “green” to the color obtained from the mixture of ocher and black, homogenizing it to the green hues created with green pigments.

### 4.3. Yellow, Red and Brown Hues

These colors are obtained using iron oxide pigments containing different kinds of iron and manganese compounds [4,51]. Iron oxide pigments have been used for centuries, from ancient times to the present, thanks to their great availability from natural sources [51]. They are widely diffused on the wall paintings of Santa Maria Antiqua and were used in all examined areas for their compatibility with fresco techniques and for their stability over time. The chromatic coordinates of iron oxide pigments are highly variable depending on their source, on the materials present in the mineral, and on their possible use mixed with other pigments to make yellow, red and brown colors lighter or darker. The values of a* and b* obtained in our measurements for yellows, reds and browns are always lower with respect to those reported in the literature [51], probably due to the presence of a surface layer of calcium carbonate white (whitening) formed because of re-crystallization phenomena.

### 4.4. Black and White Hues

Both black and white colors are present in all examined paintings and are used both to create white and black areas and mixed with other colors to produce various hues, such as the green background in the St. Anne area described earlier.

Moreover, white is added to other colors to lighten them, while black is added to darken them.

The white color is obtained with calcium carbonate [4]. In the wall paintings, generally, this pigment was used in the form of the so-called *Bianco San Giovanni*, a stable material and the same constituent of the plaster and binder for pigments in the fresco technique [52,53,54].

## 5. Conclusions

In this contribution, we report the results of the color measurements in some selected paintings of the medieval church of Santa Maria Antiqua in the Roman Forum, coupled with the multispectral similarity mapping of the same colors. An extensive color measurement campaign was never performed in Santa Maria Antiqua, so this paper contributes to better knowledge of this extraordinary challenge and the association of the observed colors with the pigments characterized in previous diagnostic investigations.

Mapping through hypercolorimetric multispectral imaging has been demonstrated as a highly useful tool for gathering the distribution of colors. Moreover, the application of this imaging technique has allowed for the acquisition of calibrated images that could be useful in the future to perform other processing, taking advantage of the PickVewer^®^ software (Version 1.0).

Thanks to careful mapping, it has been possible to identify the main hues in the investigated scenes of the wall paintings. Blues and gray hues were obtained rarely with real blue pigments but often with wine black mixed with calcium carbonate white, taking advantage of the characteristic bluish tone of this black pigment. The green color was also obtained with two different materials: green earth and a mixture of yellow ochre and carbon black. Yellow, red and brown hues were created with iron oxide pigments characterized by a variety of chromatic coordinates that depend on the source of the minerals from which the pigments were derived. Lastly, white and black hues are always present in the examined scenes. They are used both as pure colors and in mixtures with other colors to lighten or darken the hues.

In the framework of the EHEM project, the chromatic and mapping data will be useful in an attempt to reconstruct the paintings in order to enhance their appearance for the visitors to the church. In fact, the fragmentary nature of the paintings may prevent a clear reading of the pictorial scenes represented in the various areas of the church and in the different historical phases.

The future goal of the project is to use our data to improve their readability.

## Figures and Tables

**Figure 1 jimaging-10-00159-f001:**
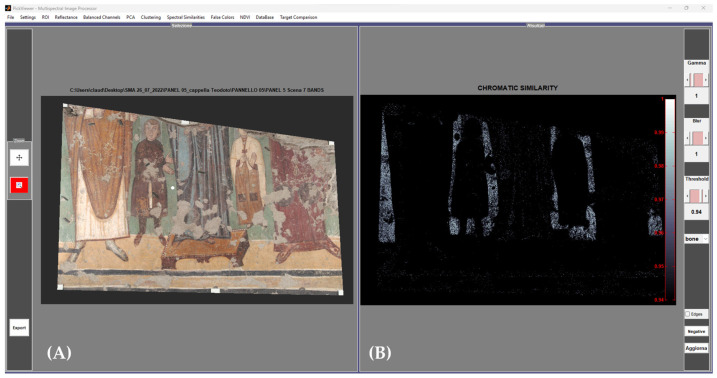
Graphical user interface of the PickViewer^®^ software (Version 1.0). (**A**) The calibrated RGB image of the painted area (Theodotus Chapel) with the point (white dot) selected for the mapping. (**B**) Result of the chromatic similarity algorithm, where the white pixels have similar color data, and the black ones have no similarity with the selected green color of the background.

**Figure 2 jimaging-10-00159-f002:**
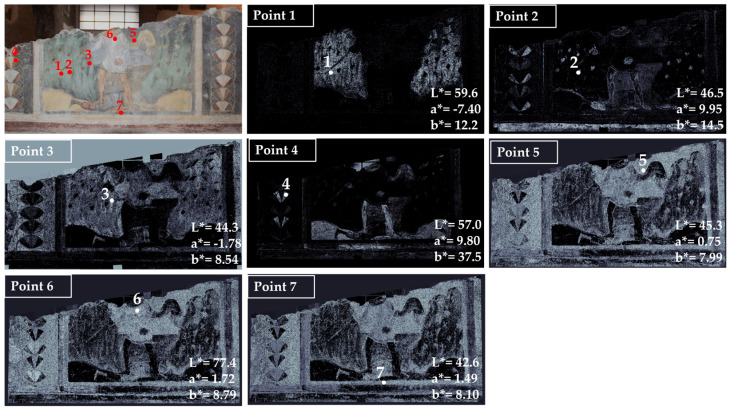
Summary of the chromatic similarity mapping and spot color measurements on area 1 (high choir, west wall). The single points are also reported in the similarity maps.

**Figure 3 jimaging-10-00159-f003:**
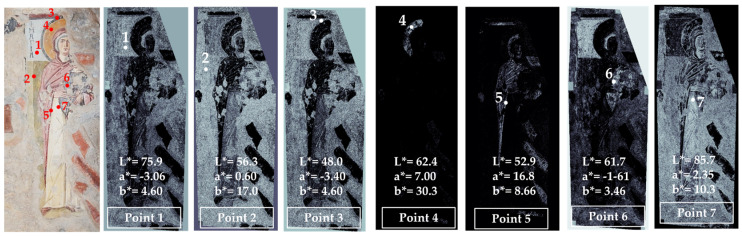
Summary of the chromatic similarity mapping and spot color measurements on area 2 (St. Anne and Child Mary). The single points are also reported in the similarity maps.

**Figure 4 jimaging-10-00159-f004:**
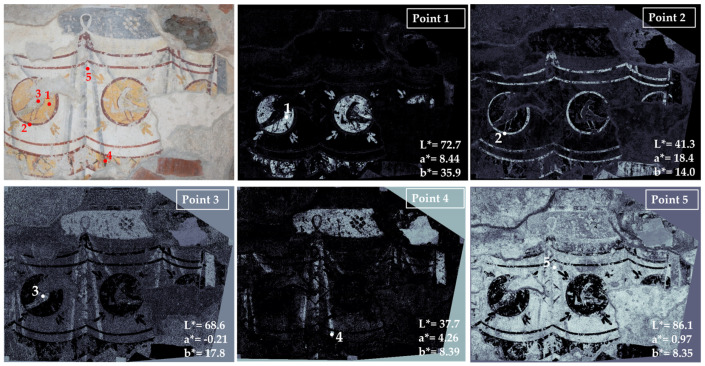
Summary of the chromatic similarity mapping and spot color measurements on area 3 (left wall with respect to the central apse). The single points are also reported in the similarity maps.

**Figure 5 jimaging-10-00159-f005:**
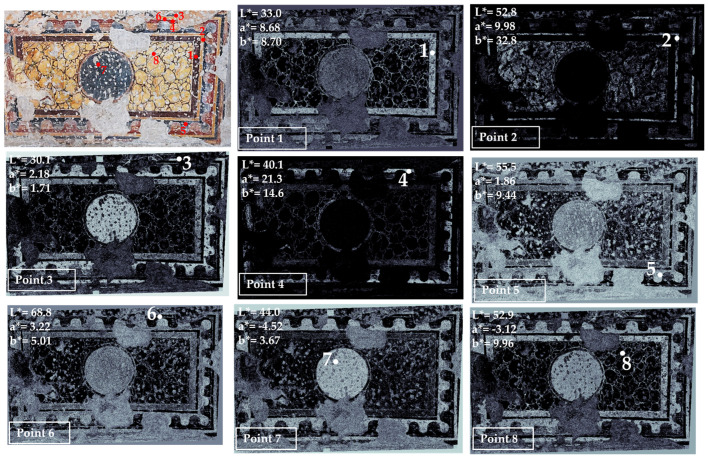
Summary of the chromatic similarity mapping and spot color measurements on area 4 (left wall with respect to the central apse), false marble decoration. The single points are also reported in the similarity maps.

**Figure 6 jimaging-10-00159-f006:**
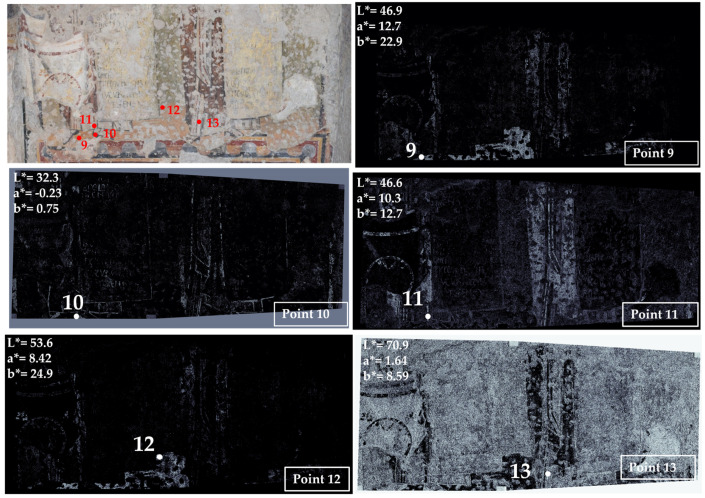
Summary of the chromatic similarity mapping and spot color measurements on area 4 (left wall with respect to the central apse), the 663 AD pictorial phase. The single points are also reported in the similarity maps.

**Figure 7 jimaging-10-00159-f007:**
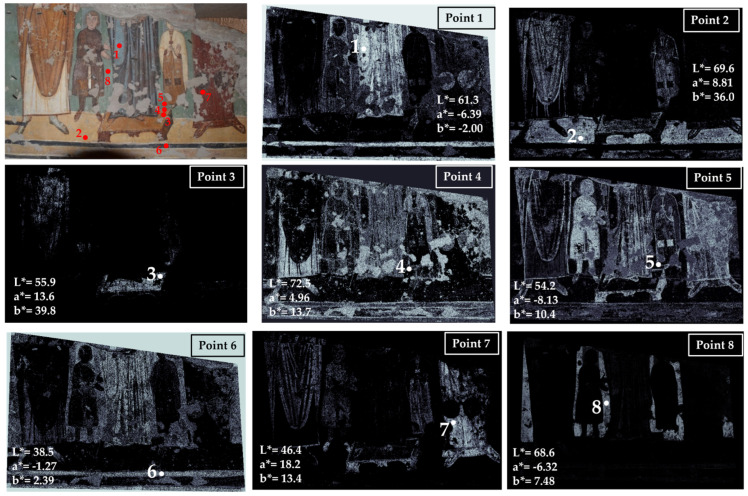
Summary of the chromatic similarity mapping and spot color measurements on area 5 in the Theodotus Chapel. The single points are also reported in the similarity maps.

**Figure 8 jimaging-10-00159-f008:**
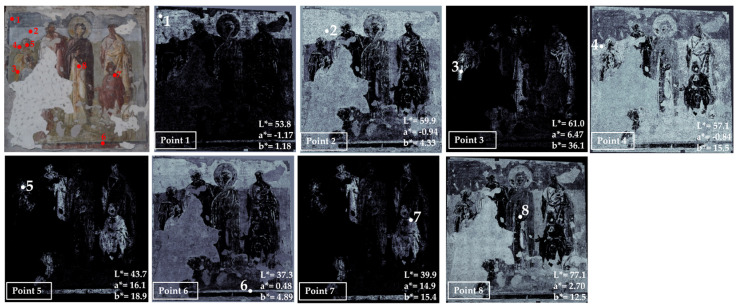
Summary of the chromatic similarity mapping and spot color measurements on area 6, the scene of Solomon and the Maccabees. The single points are also reported in the similarity maps.

**Figure 9 jimaging-10-00159-f009:**
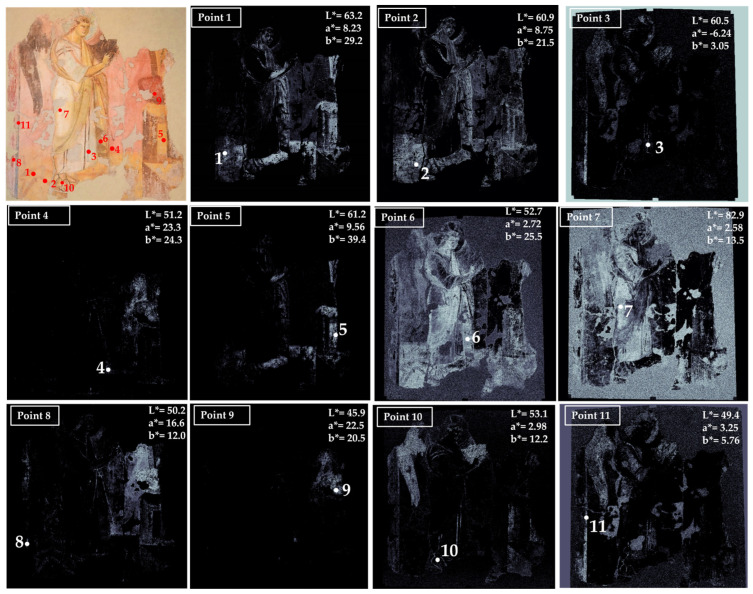
Summary of the chromatic similarity mapping and spot color measurements on area 6, the scene of the Annunciation, the pictorial phase of Pope John VII. The single points are also reported in the similarity maps.

**Figure 10 jimaging-10-00159-f010:**
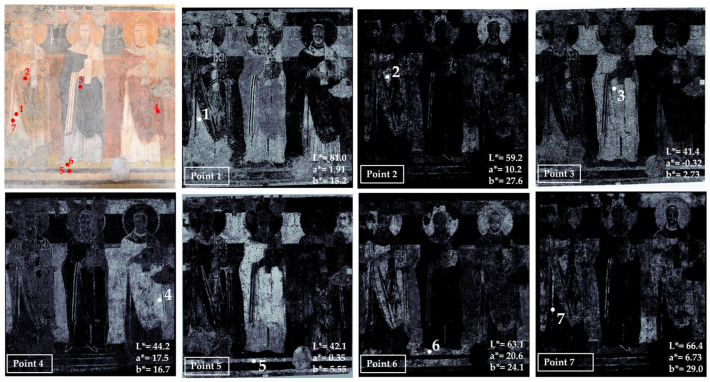
Summary of the chromatic similarity mapping and spot color measurements on area 7, the pictorial phase of Pope Stephen II. The single points are also reported in the similarity maps.

**Figure 11 jimaging-10-00159-f011:**
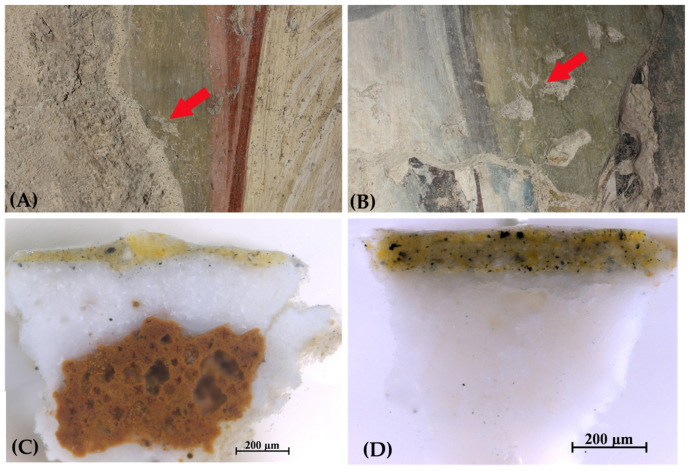
The green color obtained by mixing yellow ochre and vine black. (**A**) Image of the sampling point in the area of *Angelo Bello*; (**B**) image of the sampling point in the area of St. Anne and Child Mary; (**C**) cross-section of the sample taken from *Angelo Bello*; (**D**) cross-section of the sample taken from St. Anne. The red arrows in (**A**) and (**B**) indicate the exact points from which the samples, shown in (**C**) and (**D**) respectively, were taken.

## Data Availability

Data is available upon reasonable request.

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
