# Peer review of "Imaging Based Techniques Combined with Color Measurements for the Enhancement of Medieval Wall Paintings in the Framework of EHEM Project"

_2313-433X, 2024, doi:10.3390/jimaging10070159_

Round 1
Reviewer 1 Report
Comments and Suggestions for Authors
General comments:
The manuscript is well written and in scope of this journal. The tested HMI with spot colour measurements gives highly desirable results and is very capable. Result presentation is incomplete, however it does support the conclusions. The obtained results are discussed and cited appropriately. English corrections are needed.
Specific comments:
Results
Figure 2. The red markings of sampling points are nearly unreadable, please improve this.
Figure 3. The red markings of sampling points are nearly unreadable, please improve this.
Figure 4. The red markings of sampling points are nearly unreadable, please improve this.
Figure 5. The red markings of sampling points are nearly unreadable, please improve this.
Figure 6. The red markings of sampling points are nearly unreadable, please improve this.
Figure 7. The red markings of sampling points are nearly unreadable, please improve this.
Figure 8. The red markings of sampling points are nearly unreadable, please improve this.
Figure 9. The red markings of sampling points are nearly unreadable, please improve this.
Figure 10. The red markings of sampling points are nearly unreadable, please improve this.
This goes for figures 2-10: on each chromatic similarity map (frame) mark the sampling point location of the corresponding spot colour measurement.
Grammatical corrections:
Abstract
Better to write: 2) Methods: Two methods were employed to gather information about colors and mapping. Specifically, colorimetry was utilized for spot measurements, and hypercolorimetric multispectral imaging (HMI) was employed to map the same colors sampled through colorimetry.
Better to write: Additionally, chromatic similarity maps were generated using the innovative HMI system, a multispectral imaging technique capable of obtaining color data information through advanced calibration software named SpectraPick®. This comprehensive approach facilitated a thorough understanding of the color characteristics and distribution.
Better to write: The color measurements and mapping represent significant advancements in the interpretation of Medieval wall paintings, which are often fragmentary and stratigraphically complex.
Better to write: This research sheds new light on the colors used and enhances our understanding of the original appearance of the iconographic patterns.
Better to write: Furthermore, it enables the reconstruction of colors that closely resemble the originals.
Introduction
Better to write: This digital representation of the real monument integrates information from different sources, visualizes the monument through the various transformations it has undergone over the centuries, and allows for simulations and analyses.
Better to write: Overall, there are around twenty decorative phases, with the fulcrum of early medieval pictorial evidence found on the wall to the right of the apse (the so-called "palimpsest wall"), which features eight layers of wall paintings executed between the 4th and 8th centuries.
Better to write: The state of conservation is a result of the paintings being buried until their discovery and the subsequent changes they underwent, particularly the whitening of the surface when exposed to air and high relative humidity.
Better to write: A similar investigation was conducted on Giotto's paintings, although with a different approach (5).
Better to write: Here we propose, for the first time, the combined use of an innovative technique named Hypercolorimetric Multispectral Imaging (HMI), coupled with colorimetry, for mapping the colors, thus allowing the images to be "returned" for their digital reproduction.
Better to write: Multispectral imaging investigations enable us to deepen our knowledge of the execution technique and the state of conservation of the paintings, while also adding valuable information for defining color, especially where it is present only in traces.
Better to write: Add reference here: As demonstrated in the literature, imaging techniques have become today widely applied in cultural heritage because they can provide a complete knowledge of the surfaces in a fast and reliable way without the need for sampling micro-chips of materials for laboratory analysis, a fact that is in general highly desirable, or even mandatory in some cases [6-12; Sandak, Jakub, et al. "Nondestructive evaluation of heritage object coatings with four hyperspectral imaging systems." Coatings 11.2 (2021): 244.. ]
Discussion:
Better to write: However, until now, a comprehensive color measurement campaign had never been performed. For this reason, the EHEM project focused on the evaluation and mapping of colors.
Better to write: Previous analyses of various samples with blue or bluish appearances revealed two kinds of materials used to achieve the blue appearance: Egyptian blue and wine black [4, 29-31].
Better to write: The study of constituent materials has revealed the widespread use of earth green pigment, which is often darkened with carbon black or lightened with lime white. Additionally, there is evidence of a less common green hue obtained by mixing yellow ochre with carbon black (Figure 11).
Better to write: Both black and white colors are present in all examined paintings and are used both to create white and black areas and mixed with other colors to produce various hues, such as the green background in the St. Anne area described earlier.
Better to write: Moreover, white is added to other colors to lighten them, while black is added to darken them.
Comments on the Quality of English Language/
Author Response
General comments:
The manuscript is well written and in scope of this journal. The tested HMI with spot colour measurements gives highly desirable results and is very capable. Result presentation is incomplete, however it does support the conclusions. The obtained results are discussed and cited appropriately. English corrections are needed.
Authors reply: we thank the reviewer for the positive evaluation of our paper and for the detailed comments. The suggestions given are very useful to improve the quality of the manuscript.
Corrections and additions in the manuscript are highlighted in red characters to make them immediately visible.
Specific comments:
Results
Figure 2. The red markings of sampling points are nearly unreadable, please improve this.
Figure 3. The red markings of sampling points are nearly unreadable, please improve this.
Figure 4. The red markings of sampling points are nearly unreadable, please improve this.
Figure 5. The red markings of sampling points are nearly unreadable, please improve this.
Figure 6. The red markings of sampling points are nearly unreadable, please improve this.
Figure 7. The red markings of sampling points are nearly unreadable, please improve this.
Figure 8. The red markings of sampling points are nearly unreadable, please improve this.
Figure 9. The red markings of sampling points are nearly unreadable, please improve this.
Figure 10. The red markings of sampling points are nearly unreadable, please improve this.
This goes for figures 2-10: on each chromatic similarity map (frame) mark the sampling point location of the corresponding spot colour measurement.
Authors reply: we thank the reviewer for the comment. To make the points on the RGB image visible we reported the visible images in the supplementary. But, if the reviewer thinks that it would be better to shows the images in the main text we can move them from the supplementary to the manuscript. We chose to use supplementary in order to do not weight down the document.
Concerning the chromatic similarity map, we added the sampling point location of the corresponding spot colour measurement, as suggested by the reviewer.
Grammatical corrections:
Abstract
Better to write: 2) Methods: Two methods were employed to gather information about colors and mapping. Specifically, colorimetry was utilized for spot measurements, and hypercolorimetric multispectral imaging (HMI) was employed to map the same colors sampled through colorimetry.
Authors reply: we thank the reviewer for the comment. Text was modified as indicated
Better to write: Additionally, chromatic similarity maps were generated using the innovative HMI system, a multispectral imaging technique capable of obtaining color data information through advanced calibration software named SpectraPick®. This comprehensive approach facilitated a thorough understanding of the color characteristics and distribution.
Authors reply: we thank the reviewer for the comment. Text in the abstract has been modified as suggested by the reviewer.
Better to write: The color measurements and mapping represent significant advancements in the interpretation of Medieval wall paintings, which are often fragmentary and stratigraphically complex.
Authors reply: we thank the reviewer for the comment. Text in the abstract has been modified as suggested by the reviewer.
Better to write: This research sheds new light on the colors used and enhances our understanding of the original appearance of the iconographic patterns.
Authors reply: we thank the reviewer for the comment. Text in the abstract has been modified as suggested by the reviewer.
Better to write: Furthermore, it enables the reconstruction of colors that closely resemble the originals.
Authors reply: we thank the reviewer for the comment. Text in the abstract has been modified as suggested by the reviewer
Introduction
Better to write: This digital representation of the real monument integrates information from different sources, visualizes the monument through the various transformations it has undergone over the centuries, and allows for simulations and analyses.
Authors reply: we thank the reviewer for the comment. Text in the introduction has been modified as suggested by the reviewer
Better to write: Overall, there are around twenty decorative phases, with the fulcrum of early medieval pictorial evidence found on the wall to the right of the apse (the so-called "palimpsest wall"), which features eight layers of wall paintings executed between the 4th and 8th centuries.
Authors reply: we thank the reviewer for the comment. Text in the introduction has been modified as suggested by the reviewer
Better to write: The state of conservation is a result of the paintings being buried until their discovery and the subsequent changes they underwent, particularly the whitening of the surface when exposed to air and high relative humidity.
Authors reply: we thank the reviewer for the comment. Text in the introduction has been modified as suggested by the reviewer
Better to write: A similar investigation was conducted on Giotto's paintings, although with a different approach (5).
Authors reply: we thank the reviewer for the comment. Text in the introduction has been modified as suggested by the reviewer
Better to write: Here we propose, for the first time, the combined use of an innovative technique named Hypercolorimetric Multispectral Imaging (HMI), coupled with colorimetry, for mapping the colors, thus allowing the images to be "returned" for their digital reproduction.
Authors reply: we thank the reviewer for the comment. Text in the introduction has been modified as suggested by the reviewer
Better to write: Multispectral imaging investigations enable us to deepen our knowledge of the execution technique and the state of conservation of the paintings, while also adding valuable information for defining color, especially where it is present only in traces.
Authors reply: we thank the reviewer for the comment. Text in the introduction has been modified as suggested by the reviewer
Better to write: Add reference here: As demonstrated in the literature, imaging techniques have become today widely applied in cultural heritage because they can provide a complete knowledge of the surfaces in a fast and reliable way without the need for sampling micro-chips of materials for laboratory analysis, a fact that is in general highly desirable, or even mandatory in some cases [6-12; Sandak, Jakub, et al. "Nondestructive evaluation of heritage object coatings with four hyperspectral imaging systems." Coatings 11.2 (2021): 244.. ]
Authors reply: we thank the reviewer for the comment. Text in the introduction has been modified as suggested by the reviewer. The reference has been added, it is very interesting and suitable in this point. The subsequent references had been re-numbered.
Discussion:
Better to write: However, until now, a comprehensive color measurement campaign had never been performed. For this reason, the EHEM project focused on the evaluation and mapping of colors.
Authors reply: we thank the reviewer for the comment. Text has been modified as suggested by the reviewer.
Better to write: Previous analyses of various samples with blue or bluish appearances revealed two kinds of materials used to achieve the blue appearance: Egyptian blue and wine black [4, 29-31].
Authors reply: we thank the reviewer for the comment. Text has been modified as suggested by the reviewer.
Better to write: The study of constituent materials has revealed the widespread use of earth green pigment, which is often darkened with carbon black or lightened with lime white. Additionally, there is evidence of a less common green hue obtained by mixing yellow ochre with carbon black (Figure 11).
Authors reply: we thank the reviewer for the comment. Text has been modified as suggested by the reviewer.
Better to write: Both black and white colors are present in all examined paintings and are used both to create white and black areas and mixed with other colors to produce various hues, such as the green background in the St. Anne area described earlier.
Authors reply: we thank the reviewer for the comment. Text has been modified as suggested by the reviewer.
Better to write: Moreover, white is added to other colors to lighten them, while black is added to darken them.
Authors reply: we thank the reviewer for the comment. Text has been modified as suggested by the reviewer.
Reviewer 2 Report
Comments and Suggestions for Authors
Typo: 2.1 Colorimetry should be 2.2 Colorimetry
- I would suggest to add "Related Works" to add some contexts of what have been done with similar works in the past.
- Add more description of "Colorimetry" in section 2.2. Current description is too short.
- What would be the state-of-the-art methods for similar task? If possible, compare the method used in this study with the state-of-the-art methods.
Author Response
Typo: 2.1 Colorimetry should be 2.2 Colorimetry
Authors reply: we thank the reviewer for the comment. The subparagraph numbering has been corrected.
- I would suggest to add "Related Works" to add some contexts of what have been done with similar works in the past.
Authors reply: this comment is not so clear for us. As written in the introduction, our approach, based on the HMI mapping and point color measurements, is absolutely new and no one has applied this method before, mainly because HMI is an innovative hypercolorimetric system. In the literature you can find papers on HMI from our group or Melis (the creator of HMI) group only.
Anyway, we tried to add some more information about color measurement and mapping in the introduction together with some possible related works.
- Add more description of "Colorimetry" in section 2.2. Current description is too short.
Authors reply: we thank the reviewer for the comment. We added more details in the section 2.2.
- What would be the state-of-the-art methods for similar task? If possible, compare the method used in this study with the state-of-the-art methods.
Author reply: to our knowledge there are no other works reporting an approach similar to that used in our paper. First of all, HMI is an innovative technique developed by Profilocolore. If you find in the literature the papers dealing with this technique are only those published by our group or by Marcello Melis who is the creator of HMI system. Before the present paper, no other works were made on the application of HMI combined with color measurements to map painting areas: this was an innovative approach developed specifically for the requirements of the EHEM project.
Papers dealing with color measurements are numerous in the literature, but no one uses an approach similar to that used in our paper. Generally, color measurements are used to evaluate difference in chromatic coordinates because of a surface treatment, for example artificial ageing, cleaning, laser treatment, application of coatings, etc. or for documentation puroposes.
Anyway, we tried to add some more information about color measurements and possible related papers.
We hope to have satisfied the reviewer request.
Corrections and additions in the manuscript are highlighted in red characters to make them immediately visible.
Reviewer 3 Report
Comments and Suggestions for Authors
The paper entitled “Imaging based techniques combined with color measurements 2 for the enhancement of Medieval wall paintings in the frame-3 work of EHEM project” present a colorimetric spot measurement and hypercolorimetric multispectral imaging (HMI) approach to study and map the colors of the Medieval wall painting of three historical sites.
The work proposes a useful and easy-to-use procedure applied to several wall paintings providing an effective characterization of the relative palettes. The language and the general reading of the work is fluent, the text rather appropriately contextualized with the literature and adequately discussed.
In my opinion, the paper is suitable to be published in Journal of Imaging after minor revisions.
Suggestions and corrections are following:
Introduction
Pag 1, line 40: please substitute “Santa Maria Antiqua in Rome” with “Santa Maria Antiqua (Rome)”.
Pag 3, line 102-108: I would suggest reporting a summary of the paper less schematic, not subdivided per paragraphs but more in a discursive way.
Material and methods
2.1 The paintings chosen to apply the color measurement and mapping methodology:
I would suggest a different title “Selected paintings and mapping methodology”.
Pag 3, line 140: please substitute “scene was” with “scene were”.
Discussion
Pag 12, line 408: please substitute “The wall paintings were realized” with “The wall paintings were made”.
Conclusions
Despite the color palette is described systematically in the discussion section, a very brief summary of the identified hues could be reported in this section.
Author Response
The paper entitled “Imaging based techniques combined with color measurements 2 for the enhancement of Medieval wall paintings in the frame-3 work of EHEM project” present a colorimetric spot measurement and hypercolorimetric multispectral imaging (HMI) approach to study and map the colors of the Medieval wall painting of three historical sites.
The work proposes a useful and easy-to-use procedure applied to several wall paintings providing an effective characterization of the relative palettes. The language and the general reading of the work is fluent, the text rather appropriately contextualized with the literature and adequately discussed.
In my opinion, the paper is suitable to be published in Journal of Imaging after minor revisions.
Authors reply: we thank the reviewer for the positive evaluation of our paper and for the detailed comments. The suggestions given are very useful to improve the quality of the manuscript.
Corrections and additions in the manuscript are highlighted in red characters to make them immediately visible.
Suggestions and corrections are following:
Introduction
Pag 1, line 40: please substitute “Santa Maria Antiqua in Rome” with “Santa Maria Antiqua (Rome)”.
Authors reply: we thank the reviewer for the comment. The text was corrected.
Pag 3, line 102-108: I would suggest reporting a summary of the paper less schematic, not subdivided per paragraphs but more in a discursive way.
Authors reply: we thank the reviewer for the comment. The text was corrected according to the reviewer comment.
Material and methods
2.1 The paintings chosen to apply the color measurement and mapping methodology:
I would suggest a different title “Selected paintings and mapping methodology”.
Authors reply: we thank the reviewer for the comment. The text was corrected.
Pag 3, line 140: please substitute “scene was” with “scene were”.
Authors reply: scene is singular. The correct verbal form is was.
Discussion
Pag 12, line 408: please substitute “The wall paintings were realized” with “The wall paintings were made”.
Authors reply: we thank the reviewer for the comment. The text was corrected.
Conclusions
Despite the color palette is described systematically in the discussion section, a very brief summary of the identified hues could be reported in this section.
Authors reply: we thank the reviewer for the comment. The conclusion has been modified and summary has been added as requested by the reviewer.